# Automatic 3D/2D Deformable Registration in Minimally Invasive Liver Resection using a Mesh Recovery Network

**Mathieu Labrunie**[1,2]                                    MATHIEU.LABRUNIE@ETU.UCA.FR

[1]*Université Clermont Auvergne, Clermont Auvergne INP, CHU Clermont-Ferrand, CNRS, Institut Pascal, F-63000, Clermont-Ferrand, France*

[2]*SurgAR, 22 Allée Alan Turing, 63000 Clermont-Ferrand, France*

**Daniel Pizarro**[1,3]                                    DANI.PIZARRO@GMAIL.COM

[3]*Department of Electronics, Universidad de Alcalá, 28943 Alcalá de Henares, Spain*

**Christophe Tilmant**[1]                                    CHRISTOPHE.TILMANT@UCA.FR

**Adrien Bartoli**[1,4]                                    ADRIEN.BARTOLI@GMAIL.COM

[4]*Department of Clinical Research and Innovation, CHU Clermont-Ferrand, Clermont-Ferrand, France*

**Editors:** Accepted for publication at MIDL 2023

## Abstract

We propose the patient-specific Liver Mesh Recovery (LMR) framework, to automatically achieve Augmented Reality (AR) guidance by registering a preoperative 3D model in Minimally Invasive Liver Resection (MILR). Existing methods solve registration in MILR by pose estimation followed with numerical optimisation and suffer from a prohibitive intraoperative runtime. The proposed LMR is inspired by the recent Human Mesh Recovery (HMR) framework and forms the first learning-based method to solve registration in MILR. In contrast to existing methods, the computation load in LMR occurs preoperatively, at training time. We construct a patient-specific deformation model and generate patient-specific training data reproducing the typical defects of the automatically detected registration primitives. Experimental results show that LMR's registration accuracy is on par with optimisation-based methods, whilst running in real-time intraoperatively.

**Keywords:** Mini-invasive surgery, Liver, 3D/2D registration, Augmented Reality

## 1. Introduction

Minimally Invasive Liver Resection (MILR) is the surgical removal of liver tumours by laparoscopy or robot-assisted intervention. It has tremendous advantages over open surgery, reducing recovery time and the risk of postoperative complications. However, the intraoperative tumour localisation remains extremely challenging, for two main reasons. First, the mini-invasive setup prevents the surgeon to palpate the liver. Second, the liver tissue is soft, causing strong deformation under pneumoperitoneum, breathing, and mobilisation. Importantly, a preoperative volume (an abdominal CT or MRI), primarily used for diagnosis, is routinely available which clearly shows the tumours. It is however not possible to mentally transfer the tumour locations to the intraoperative field with sufficient accuracy (Adballah et al., 2022). This has motivated the development of computer assistance using Augmented Reality (AR), which combines the preoperative volume with the intraoperative images in a virtual see-through manner to reveal the tumour locations. The key problem of transferring information from the preoperative volume to the intraoperative images must thus be solved by a computational method, which currently represents the main challenge to

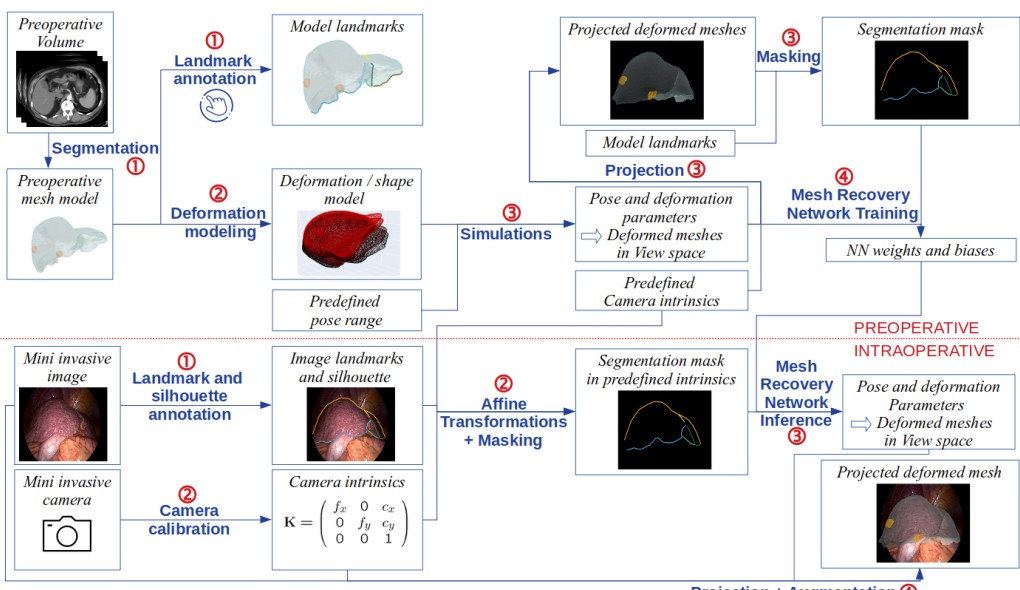

Figure 1: Proposed registration pipeline in MILR (see section 3). (top) Preoperative steps: (1) reconstruct the liver 3D model from the CT scan with anatomical landmarks, (2,3) synthesise deformations and the landmark and silhouette masks, and (4) train LMR. (bottom) Intraoperative steps: (1) segment the surgical image to recover the landmark and silhouette masks, (2) compensate the intrinsic camera parameters, (3) infer the registration with LMR, and (4) augment the image.

achieve AR in MILR. Specifically, the key problem is called *registration*, which establishes a non-rigid transformation from the preoperative volume to the intraoperative images, generally decomposed into rigid and non-rigid parts, respectively the pose and the deformation. Existing registration methods in MILR (Özgür et al., 2018; Koo et al., 2022; Labrunie et al., 2022) rely on landmark correspondences, namely the lower ridge and the falciform ligament (appendix figures 3 and 4). In the preoperative stage, the liver volume is segmented to reconstruct a virtual preoperative 3D model including the tumours and the landmarks. In the intraoperative stage, the current image is processed to detect the landmarks and the liver silhouette, called the primitives. The pose and the deformation are then computed from the primitives and biomechanical constraints. Existing methods mainly carry out the computation using nonconvex numerical optimisation, typically using a gradient-based method such as Gauss-Newton. Together, they thus form the *optimisation-based approach* to registration in MILR. The state-of-the-art accuracy measured by tumour Target Registration Error (TRE) is typically within a few centimeters, which was shown to be clinically relevant (Zhong et al., 2017). However, these methods have a main structural limitation, which is that most of the computation is done intraoperatively, leading to a prohibitive intraoperative runtime. Indeed, the intraoperative optimisation process involves an order

of 100 parameters and a nonconvex cost function, typically requiring dozens of iterations to convergence. This strongly limits the usability of these methods to achieve AR in MILR.

We propose the Liver Mesh Recovery (LMR) framework, forming a novel *learning-based approach* to registration in MILR. LMR is inspired by recent methods to reconstruct the human body as a 3D mesh from a single image (Kanazawa et al., 2018; Tian et al., 2022). LMR directly regresses the registration from the intraoperative image primitives. Specifically, the registration is represented by pose and deformation parameters that define the intraoperative liver state. A major difficulty is the absence of labelled patient-generic datasets and the fact that the real patient-specific images only become available at the time of surgery. Training from these images is thus not possible, first, as the registration labels cannot be reliably found and, second, as surgery has already started. A related difficulty is that the intrinsic camera parameters also become available only when surgery starts. We propose a novel key idea, which is to train preoperatively from patient-specific semi-synthetic data generated from the preoperative 3D model. We bring three main technical contributions. First, the LMR framework, including a learned iterative refinement loop. Second, a physics-based rendering method which generates patient-specific image primitive masks for a range of pose and deformation incorporating the typical defects of automatic primitive detectors. Third, a process to train LMR with default intrinsic camera parameters and the adaptation to their actual value at inference time. In contrast to the optimisation-based approach, the proposed LMR transfers most of the computational load to the preoperative stage, making intraoperative registration fast and fully automatic. We report experimental results showing that LMR's accuracy is on par with the state-of-the-art, thus clinically relevant, while running in real-time intraoperatively.

## 2. Related Work

### 2.1. Registration in Minimally Invasive Liver Resection

Existing methods use the same landmarks as (Plantefeve et al., 2014) and the silhouette (Koo et al., 2017). The methods generally follow two steps, shown in appendix figure 4: 1) pose estimation, forming an initial rigid registration, and 2) deformation estimation. Pose estimation requires manual interaction (Espinel et al., 2020; Özgür et al., 2018), though recent work attempted automatic solutions. The method in (Robu et al., 2018) uses a shape-matching technique with the ridge and a 3D point cloud reconstructed from stereo. The method in (Koo et al., 2022) samples the pose space and evaluates the reprojection error in the ridge and the silhouette. The method in (Labrunie et al., 2022) searches for the pose directly from the primitives using a visibility reasoning to determine landmark correspondences. The deformation is estimated by means of intraoperative numerical optimisation of a compound cost function, with a main term exploiting the primitives. Optional terms were tried, such as shading (Koo et al., 2017). The deformation is regularised from the strain energy of a constitutive model and generic biomechanical parameters. Low-dimensional deformation models such as Locally Linear Embedding (LLE) (Roweis and Saul, 2000) were also incorporated to stabilise and speed up optimisation (Modrzejewski, 2020; Labrunie et al., 2022).

## 2.2. Human Body Shape Reconstruction

Human body shape reconstruction uses a single image to estimate a physically plausible mesh aligned with the image. Most methods combine a reference shape model and joint pose (Loper et al., 2015) to model the mesh. Computationally, both the optimisation-based and learning-based approaches were studied (Tian et al., 2022). Optimisation-based methods first compute an initial pose and shape, followed by an optimisation of the shape parameters for multiple image cues, including the silhouette (Guan et al., 2009). Learning-based methods are recent and stem from the end-to-end Human Mesh Recovery (HMR) framework (Kanazawa et al., 2018). HMR starts with a ResNet-50 encoder to extract features. It then concatenates the features with the pose, shape and camera parameters, which are all fed to a regression network with two fully-connected layers. The regression network uses Iterative Error Feedback (IEF) (Carreira et al., 2016) to update the parameters iteratively. At training, the image joint locations are known. The reprojection of the estimated 3D joints allows one to compute a 2D reprojection loss. When 3D Ground Truth (GT) is known, a 3D loss is computed directly on the 3D joints as well as on the parameters. In addition, a discriminator is used to improve the shape and pose plausibility, which is especially important in the absence of 3D GT.

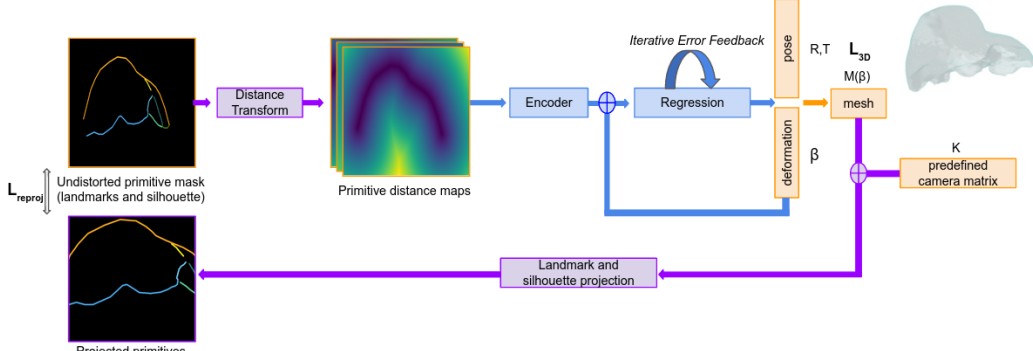

Figure 2: Proposed Liver Mesh Recovery (LMR) framework (see sections 3.1 and 3.2). The framework inputs the primitive mask for the surgical image (top left) and outputs the corresponding 3D model pose and deformation parameters (top right). It computes a distance transform, encodes it, and iteratively regresses the parameters to minimise a sum of 3D losses and the reprojection loss for the primitives.

## 3. Methodology

### 3.1. Registration Pipeline and the Liver Mesh Recovery Framework

The proposed registration pipeline is shown in figure 1 and the LMR framework in figure 2. LMR uses a ResNet-50 encoder which delivers features from the image primitive masks, represented as distance maps. The features are concatenated with the current pose and deformation parameters. They are used as inputs to the regression network, based on fully

connected layers, which iteratively updates the pose $R, T$ and deformation parameters $\beta$ through IEF (Carreira et al., 2016). We use the same number of layers and neurons in the encoder and regressor networks as in HMR (Kanazawa et al., 2018), except for the input and output layers of the regressor, adapted to the number of deformation parameters. LMR outputs the registered liver mesh, in other words, a mesh properly deformed and expressed in camera coordinates, whose reprojection matches the liver observed in the surgical image.

LMR differs from HMR in four main ways. 1) LMR uses a patient-specific biomechanical liver deformation model, in contrast to an articulated body joint model combined with a generic shape model in HMR. 2) LMR uses the pinhole camera model, which reduces depth ambiguities, in contrast to the scaled-orthographic model in HMR. 3) LMR is trained patient-specific from semi-synthetic data, as the amount of annotated data in MILR is extremely limited and the real patient-specific data only becomes available at the time of surgery, in contrast to a person-generic model in LMR trained from massive datasets. 4) LMR uses supervised training, making the adversarial discriminator from HMR unnecessary.

### 3.2. Preoperative Stage

The preoperative steps reconstruct the preoperative 3D model, which is used to synthesise images and train the LMR network.

**Step 1: preoperative 3D model reconstruction and landmark annotation.** The preoperative volume is segmented, producing a surface mesh, which is upgraded to a volume by constrained Delaunay tetrahedralisation (Shewchuk, 2002). This yields a volumetric mesh retaining the surface vertices, whose $n$ vertex coordinates, of order 10000, are stacked into the column vector $\mu^\top = [\mu_1^\top, \cdots, \mu_n^\top] \in \mathbb{R}^{3n}$. The tumours and vena cava are marked as inner regions of the volumetric mesh and the landmarks annotated on the surface. This step is done semi-automatically using standard software including 3D Slicer and Tetgen.

**Step 2: deformable liver shape modeling** In MILR, the liver mainly undergoes loads from surgical instruments and breathing. Breathing was shown to have a limited influence on the deformation compared to mobilisation (Özgür et al., 2018). We use Finite Element Analysis (FEA) to model the load incurred by a surgical instrument. We apply nodal forces on random vertices of the liver surface. We exclude vertices of the vena cava, which we use as boundary condition, as it forms the main structure maintaining the liver (Flament et al., 1982). We use the Ogden constitutive model with generic liver parameters (Pellicer-Valero et al., 2020). We implement the simulations automatically with the FEBio software (Maas et al., 2012). We preserve the mesh connectivity and thus represent each shape by its vertices in $\mathbb{R}^{3n}$. Upon using forces of random orientation and a load curve of 10 linear steps to reach the magnitude of 0.5 N, we generate $k = 5000$ simulated shapes $x_1, \ldots, x_k$.

With the aim of estimating a deformable shape model with reduced number of parameters, we follow the POD framework (Sifakis and Barbic, 2012). This uses a Truncated SVD of the synthetic shapes $x = [x_1, \cdots, x_k]^\top \in \mathbb{R}^{3n \times k}$ to construct principal components $\phi$. We keep the leading $m$ components that allow reconstructing 99% of the data within 5 mm or a minimum of 5 components. A deformed shape is thus represented by its vertex coordinates $\hat{x}_i$, obtained from its $m$ deformation coefficients $\beta_i = [\beta_{i,1}, \cdots, \beta_{i,m}]^\top$ as $\hat{x}_i = \mu + \sum_{j=1}^m \beta_{i,j}\phi_j$. Appendix figure 6 illustrates the range of the deformations induced by each model component for one patient.

**Step 3: deformation sampling and mask generation.** We complete the $k$ FEA simulated shapes to form $l \gg k$, with $l = 40000$ shapes to serve as training dataset. We fix the camera intrinsics to a default value, estimated from the laparoscope used for the first patient in our experiments. We generate $l$ camera poses by composing a default typical pose with a random pose perturbation sampled in $SE(3)$. The default typical pose puts the liver in a frontal view where its anterior ridge, ligament and silhouette are mostly visible by the camera. Specific details of the deformation sampling can be found in the appendix section B. We project the $l$ shapes using a z-buffer to handle visibility and obtain the image primitives as contours. The dataset has up to $l = 40000$ pairs of shapes and image primitives, from which samples for which fewer than two primitives are visible are removed. We split the dataset in 80% training, 10% validation and 10% test. We convert the primitive contours into segmentation masks, which we randomly perturb to emulate the typical error of automatic detection. We finally transform the masks to distance maps, using the Distance Transform (Rosenfeld and Pfaltz, 1966) which we normalise to $[-1, 1]$, with 1 associated to the image diagonal.

**Step 4: LMR training.** We use a loss with three terms. First, the reprojection loss term, which is the Mean Sum of Distances (MSD) (Li et al., 2005) between the predicted and input primitives. Second, the 3D mesh loss term, which is the Mean Absolute Error (MAE) of the euclidean distances between the predicted and GT mesh vertices. Third, the pose and deformation coefficient loss term, which is the MAE between the normalised predicted and GT pose and deformation coefficients. The last two terms are referred to as the 3D loss terms. Additional details can be found in appendix section C.

### 3.3. Intraoperative Stage

We use four intraoperative steps. Step 1 extracts the primitives automatically by means of segmentation using a COSNet (Lu et al., 2019) trained and validated as in (Labrunie et al., 2022) from 67 patients and 1373 images, from which 100 pairs per patient were constructed to serve co-attention. As the input is a single image we disable the co-attention module at inference by using the input image twice. We ran an evaluation on the annotated 46 images from the dataset (Rabbani et al., 2021). The mean MSD for the baseline (Labrunie et al., 2022) and for COSNet are 51 px and 40 px, showing an improvement. Step 2 calibrates the camera when surgery starts, giving $K_{\text{actual}}$. The parameters were unknown preoperatively; recall that default intrinsics $K_{\text{default}}$ were used instead. LMR was thus trained to handle images with different intrinsics than the ones of the actual surgical image. Step 2 copes with this difference by adapting the primitive segmentation masks prior to their use by LMR, by applying a 2D affine warp $A = K_{\text{default}} K_{\text{actual}}^{-1}$ (Fuentes-Jimenez et al., 2022). Step 3 performs registration using LMR and step 4 uses the registration to augment the image with the hidden anatomical structures transferred from the preoperative 3D model.

## 4. Experimental Results

We tested our registration framework with a publicly available clinical dataset[1] designed to evaluate registration accuracy in MILR. This dataset consists of four in-vivo patient

---

1. http://igt.ip.uca.fr/~ab/code_and_datasets/datasets/llr_reg_evaluation_by_lus

cases along with annotated preoperative and intraoperative images, camera calibration, and the facility to measure the predicted tumour Target Registration Error (TRE) with GT obtained by Laparoscopic Ultrasound (LUS) in controlled conditions (Rabbani et al., 2021). We adjusted the landmark annotations of the preoperative 3D model and the images to match the split-ridge model (Labrunie et al., 2022). We used barycentric coordinates to resample the model landmarks to reach four landmark points per mm. We trained LMR from simulations obtained with maximal force magnitudes of 0.5N and 5N. The two results are labelled as OLA (Ogden - Low Amplitude) and OHA (Ogden - High Amplitude). We conducted an evaluation on semi-synthetic and real data.

| MAE (mm) | | Patient 1 | Patient 2 | Patient 3 | Patient 4 | Average |
|---|---|---|---|---|---|---|
| Average | OLA | 7.5 | 6.1 | 11.4 | 9.0 | 8.5 |
| | OHA | 13.7 | 20.5 | 18.2 | 13.3 | 16.4 |
| $\eta_{.95}$ | OLA | 21.4 | 18.0 | 26.9 | 23.3 | 22.4 |
| | OHA | 29.0 | 42.6 | 38.9 | 29.1 | 34.9 |

Table 1: MAE average and 95th percentile on the test dataset for each patient.

### 4.1. Semi-synthetic Dataset

We use the test dataset generated along with the LMR training data, see section 3.2. We measure the Mean Absolute Error (MAE) of the euclidean distances between the estimated and GT mesh vertices, see table 1. The average MAE in each patient is lower than or close to 1 cm for OLA. 95% of the test samples have an MAE within $[18, 27]$ mm. OHA obtains a larger MAE, about twice as large as the average for OLA. Appendix figure 8 shows results for Patient 1 and table 3 for the ablation of the loss terms for OLA. Not using the 3D losses leads to slightly larger MAE while not using the reprojection loss leads to much larger MAE.

### 4.2. Real Dataset

We measure the tumour TRE of the proposed LMR for the real dataset (Rabbani et al., 2021). We compare with the following three baselines. M-Pose is the rigid pose manually given by an expert. A-Pose is a rigid pose estimated from the primitives (Labrunie et al., 2022). Opt-B is a deformation estimated by refining the result of A-Pose (Labrunie et al., 2022), using a Neo-Hookean constitutive deformation model. The methods are run from manual or automatic image annotations. The results are given in table 2. Opt-B obtains the best performance in all patients for manual annotations, with a TRE within $[9, 15]$ mm, except for Patient 2 where a strong deformation induced by the ultrasound probe leads to over 6 cm TRE. LMR mainly obtains TRE lower than 2 cm for OLA with a smaller standard deviation, and generally higher TRE and standard deviation for OHA, except for Patient 4. OLA also obtains a high TRE for Patient 2 but outperforms Opt-B. The performance of both methods degrade for automatic image annotation, owing to segmentation errors (see appendix figure 10). Appendix figure 9 shows the detected and reprojected primitives for OLA. The observed discrepancy suggests that the simulation domain does not fully span the real domain. This is confirmed by the ablation study of the loss terms for OLA on the real dataset, see appendix table 4. Indeed, unlike on the semi-synthetic test dataset, only

| TRE (mm) | Patient 1 | | Patient 2 | | Patient 3 | | Patient 4 | |
|---|---|---|---|---|---|---|---|---|
| Methods | A | M | A | M | A | M | A | M |
| M-Pose | $15.1^{\pm 2.3}$ | | $35.5^{\pm 12.2}$ | | $30.5^{\pm 5.2}$ | | $16.3^{\pm 2.3}$ | |
| A-Pose | $30.0^{\pm 10.8}$ | $9.8^{\pm 3.0}$ | $86.9^{\pm 14.7}$ | $63.0^{\pm 11.4}$ | $29.8^{\pm 60.2}$ | $9.7^{\pm 2.3}$ | $20.5^{\pm 1.0}$ | $17.8^{\pm 0.8}$ |
| Opt-B | $29.3^{\pm 10.6}$ | $10.3^{\pm 2.8}$ | $86.9^{\pm 14.7}$ | $63.0^{\pm 11.4}$ | $30.0^{\pm 60.1}$ | $9.5^{\pm 2.1}$ | $19.8^{\pm 1.0}$ | $14.7^{\pm 1.2}$ |
| LMR OLA | $17.9^{\pm 3.7}$ | $17.4^{\pm 4.2}$ | $45.6^{\pm 14.7}$ | $53.8^{\pm 8.0}$ | $46.9^{\pm 7.1}$ | $17.6^{\pm 0.6}$ | $22.7^{\pm 9.1}$ | $17.0^{\pm 0.8}$ |
| LMR OHA | $22.2^{\pm 5.1}$ | $19.7^{\pm 5.4}$ | $54.4^{\pm 15.5}$ | $63.7^{\pm 9.9}$ | $61.6^{\pm 4.8}$ | $19.3^{\pm 2.2}$ | $19.5^{\pm 4.4}$ | $14.6^{\pm 0.5}$ |

Table 2: Average tumour TRE and standard deviation ($\pm$) for the real data. Columns A and M respectively stand for automatic and manual image annotations.

using the reprojection term leads to similar TRE, even lower for Patient 4 than the full loss, while only using the 3D losses results in higher TRE. Nonetheless, the limited amount of real evaluation data does not allow us to conclude about the optimal loss function, even though a good compromise seems to be the use of both the reprojection and 3D loss terms.

All experiments were run on a medium-end Nvidia GeForce RTX 2080 GPU. The inference time was $5 \pm 3$ s for A-Pose, $43 \pm 47$ s for the optimisation in Opt-B and $3.4 \pm 0.2$ ms for LMR inference. Primitive detection with COSNet takes $9.3 \pm 0.2$ ms. This means that our pipeline allows real-time processing, with at least $16 \pm 2$ fps as obtained with our unoptimised implementation. FEA simulation and mask generation took 4 hours to run and LMR 16 hours to train per patient. This can be reduced by using a high-end GPU, but is already compatible with the typical delay between preoperative scanning and surgery.

## 5. Conclusion

We have proposed LMR, a novel patient-specific learning-based registration method in MILR. LMR directly regresses the registration parameters from primitives detected in the intraoperative image. In contrast to the state-of-the-art optimisation-based approaches, LMR transfers most of the computation to the preoperative stage, where it is trained from physics-based simulations. Inference takes a few forward passes -we use three passes-through a neural model. It is initialisation-free and fast, offering a speed up of five orders of magnitude compared to existing work. Our experimental results suggest that LMR obtains competitive results in real clinical cases, achieving a tumour TRE lower than 2 cm, except for a large deformation case defeating all existing methods. Our current implementation of LMR has three main limitations. First, training on semi-synthetic data creates a domain shift. We plan to address this issue in two ways: 1) by improving the way we simulate defects in the simulated primitive masks, and 2) by improving the proposed primitive detector (by training on a larger dataset and using multiple images for co-segmentation). Second, training on semi-synthetic data limits generalisation in terms of recoverable deformation span. We plan to address this issue by completing our physics-based simulation with additional loads modelling breathing and the pneumoperitoneum. Third, LMR is patient-specific, meaning that training or fine-tuning is required for each patient, forming a practical limitation. Making LMR patient-generic would incur profound design changes and would require the availability of a large dataset.

## Acknowledgments

M. Labrunie is supported by a CIFRE PhD fellowship (N°2021/0184) from ANRT under partnership between EnCoV and SurgAR. This research has been funded by Cancéropôle CLARA within the Proof-of-Concept project AIALO (2020-2023) and has also been supported by the Spanish Ministry of Science and Innovation MCIN/AEI/10.13039/501100011033 through the Project ATHENA under Grant PID2020-115995RB-I00.

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

## Appendix A. Ablation Study

| MAE (mm) | | Patient 1 | Patient 2 | Patient 3 | Patient 4 | Average |
|---|---|---|---|---|---|---|
| Aver. | Both | 7.5 | 6.1 | 11.4 | 9.0 | 8.5 |
| | Reproj only | 30.6 | 57.5 | 27.9 | 22. | 34.5 |
| | 3D only | 8.8 | 7.0 | 12.1 | 10.7 | 9.7 |
| $\eta_{.95}$ | Both | 21.4 | 18.0 | 26.9 | 23.3 | 22.4 |
| | Reproj only | 75.0 | 128.1 | 62.3 | 58.9 | 81.1 |
| | 3D only | 24.1 | 19.2 | 29.8 | 27.1 | 25.1 |

Table 3: MAE average 'Aver.' and 95th percentile on the test dataset for each patient in the ablation study with different losses using OLA. 'Both' (the method in green, see table 1) corresponds to the use of both reprojection and 3D losses, while 'Reproj only' and '3D only' respectively use the reprojection or the 3D losses only.

| TRE (mm) Loss terms | Patient 1 | | Patient 2 | | Patient 3 | | Patient 4 | |
|---|---|---|---|---|---|---|---|---|
| | A | M | A | M | A | M | A | M |
| Both | $17.9^{\pm3.7}$ | $17.4^{\pm4.2}$ | $45.6^{\pm14.7}$ | $53.8^{\pm8.0}$ | $46.9^{\pm7.1}$ | $17.6^{\pm0.6}$ | $22.7^{\pm9.1}$ | $17.0^{\pm0.8}$ |
| Reproj only | $22.5^{\pm5.0}$ | $17.2^{\pm2.6}$ | $46.2^{\pm22.9}$ | $51.6^{\pm4.0}$ | $53.2^{\pm6.1}$ | $17.7^{\pm0.6}$ | $8.4^{\pm3.5}$ | $8.8^{\pm2.1}$ |
| 3D only | $15.2^{\pm4.1}$ | $20.9^{\pm3.7}$ | $37.8^{\pm10.9}$ | $52.0^{\pm7.3}$ | $44.9^{\pm7.0}$ | $33.6^{\pm1.7}$ | $24.4^{\pm1.6}$ | $25.9^{\pm2.5}$ |

Table 4: Average tumour TRE and standard deviation ($\pm$) for the real data in the ablation study with different losses for OLA. 'Both' (the method in green, see table 2) corresponds to the use of both reprojection and 3D losses, while 'Reproj only' and '3D only' respectively use the reprojection or the 3D losses only. Columns A and M respectively stand for automatic and manual image annotations.

## Appendix B. Deformation Sampling Details

The extra $(k-l)$ shapes are sampled from a normal distribution using as standard deviation $\frac{2}{3}$ of the one from the initial $k$ coefficients. This guarantees that 99% of the complete set of shapes is within twice the initial standard deviation. The default typical pose uses $X, Y, Z$ rotation angles of $-65, 45, -45$ degrees and $X, Y, Z$ translations of $0, 0, 175$ mm. The random pose perturbation uses a uniform distribution on the $X, Y, Z$ rotation angles within $50, 30, 40$ degrees and on the $X, Y, Z$ translations of $40, 40, 100$ mm. We use PyTorch3D with its default coordinate system. Appendix figure 5 illustrates the above ranges for one of the patients from our experiments.

## Appendix C. Training Details

We use weights of 60, 0.1 and 1 respectively for the three terms. We use the SGD optimiser with 8 samples in the batch, 3 IEF iterations and 50 training epochs. We use the same learning rate for the encoder and regressor, fixed to 1e-3 for the first 45 epochs and then set to 1e-4 with a weight decay of 1e-4.

## Appendix D. Additional Figures

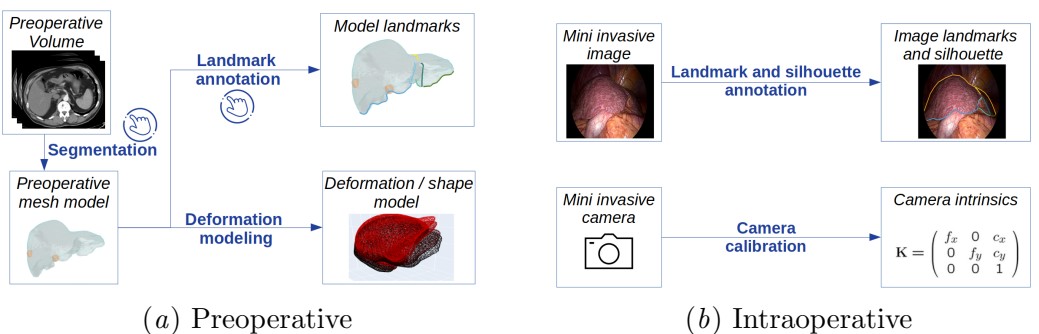

(*a*) Preoperative            (*b*) Intraoperative

Figure 3: Steps followed to solve registration in MILR, common to both existing optimisation-based methods and the proposed learning-based method.

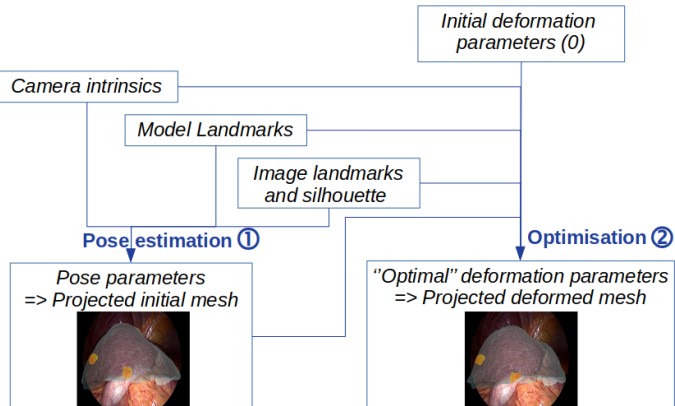

Figure 4: Steps to solve registration in MILR specific to the optimisation-based approach.

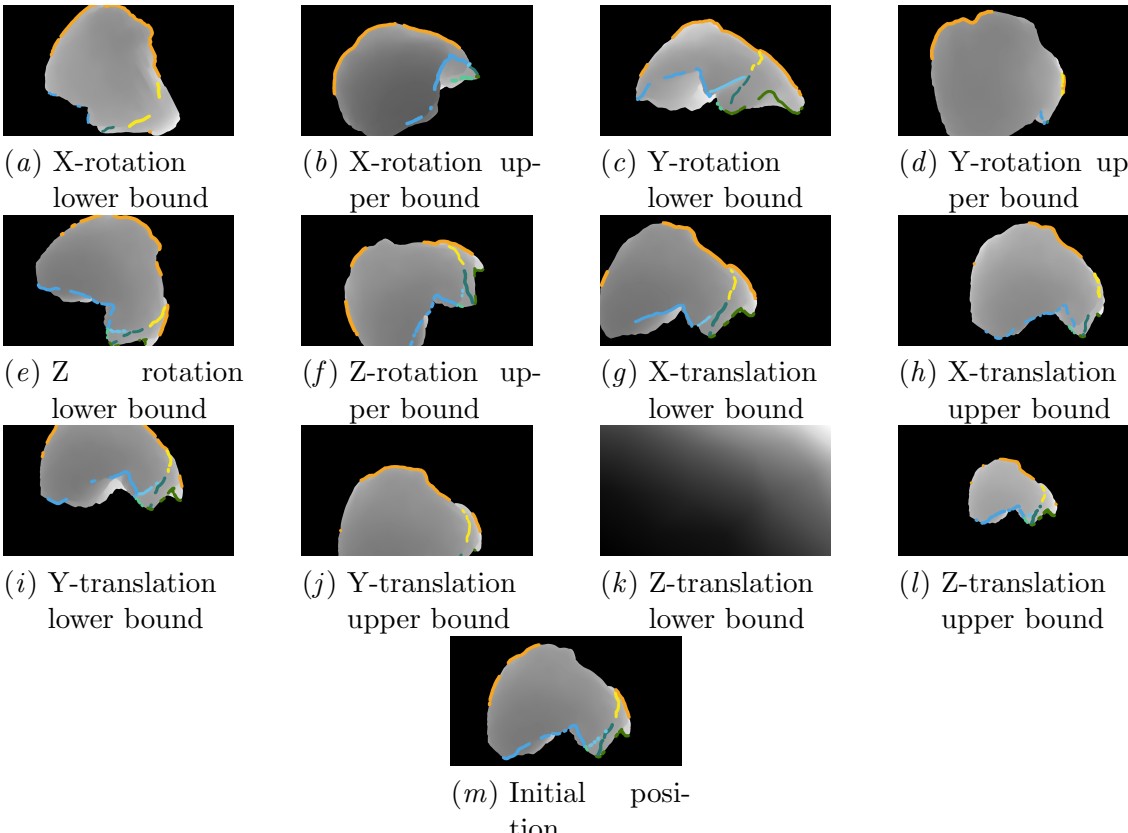

(a) X-rotation lower bound

(b) X-rotation upper bound

(c) Y-rotation lower bound

(d) Y-rotation upper bound

(e) Z rotation lower bound

(f) Z-rotation upper bound

(g) X-translation lower bound

(h) X-translation upper bound

(i) Y-translation lower bound

(j) Y-translation upper bound

(k) Z-translation lower bound

(l) Z-translation upper bound

(m) Initial position

Figure 5: Undeformed liver configurations obtained within the bounds of the pose simulation range where only one parameter is modified compared with the reference configuration (m), for Patient 4. Examples of advanced simulated primitives are superimposed to the depth map and enlarged for visualisation purposes. The silhouette, the ligament junction and the split ridge are respectively in orange, yellow and nuances of green and blue.

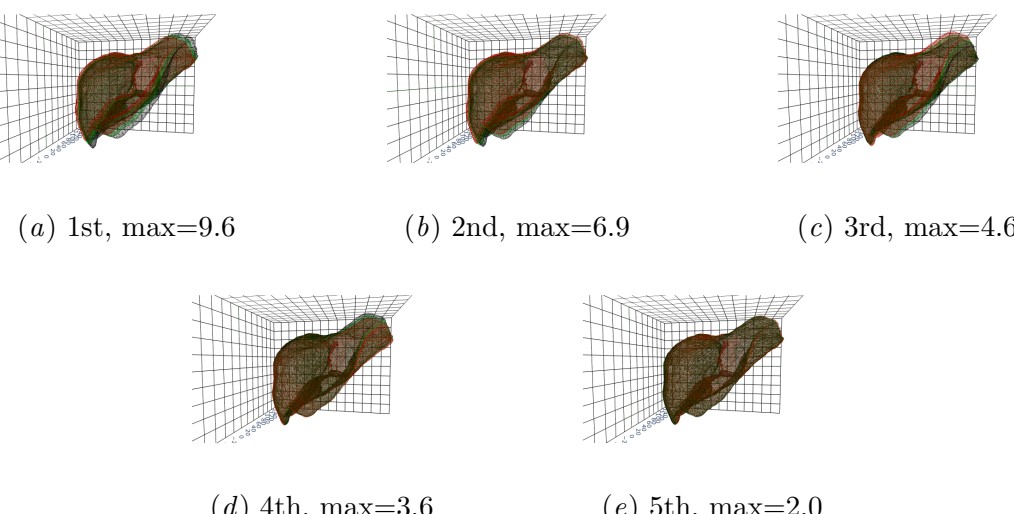

(a) 1st, max=9.6          (b) 2nd, max=6.9          (c) 3rd, max=4.6

(d) 4th, max=3.6          (e) 5th, max=2.0

Figure 6: Nomograms of the deformation components for OLA in Patient 1. In green, the initial mesh, and in red and black the deformed meshes within the bounds of the deformation ranges. Captions include the maximal vertex displacement (mm) from the rest configuration.

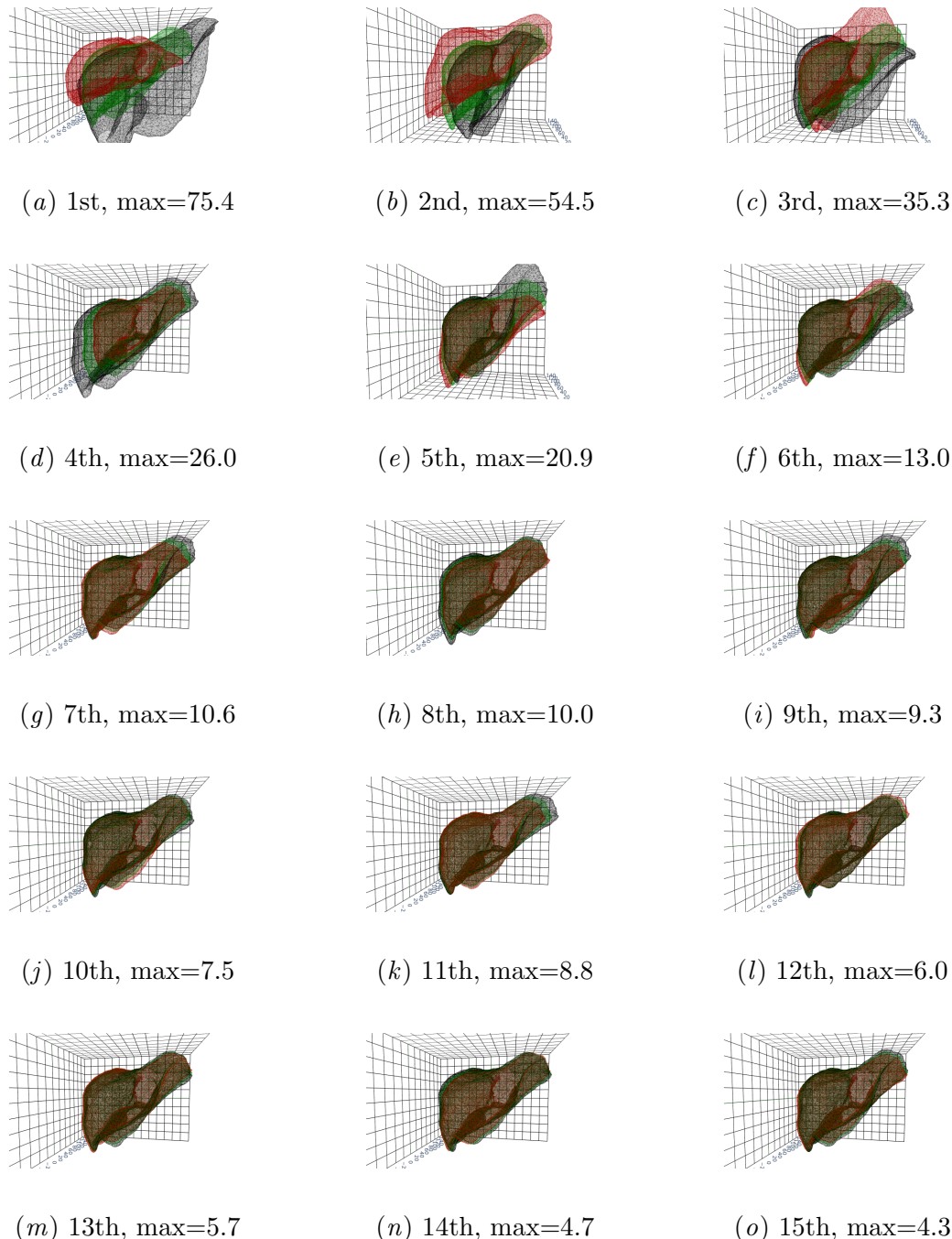

($a$) 1st, max=75.4          ($b$) 2nd, max=54.5          ($c$) 3rd, max=35.3

($d$) 4th, max=26.0          ($e$) 5th, max=20.9          ($f$) 6th, max=13.0

($g$) 7th, max=10.6          ($h$) 8th, max=10.0          ($i$) 9th, max=9.3

($j$) 10th, max=7.5          ($k$) 11th, max=8.8          ($l$) 12th, max=6.0

($m$) 13th, max=5.7          ($n$) 14th, max=4.7          ($o$) 15th, max=4.3

Figure 7: Nomograms of the first 15 deformation components (out of 20) for OHA in Patient 1. In blue, the initial mesh, and in red and black the deformed meshes within bounds of the deformation ranges. The captions include the maximal vertex displacement (mm) from the rest configuration.

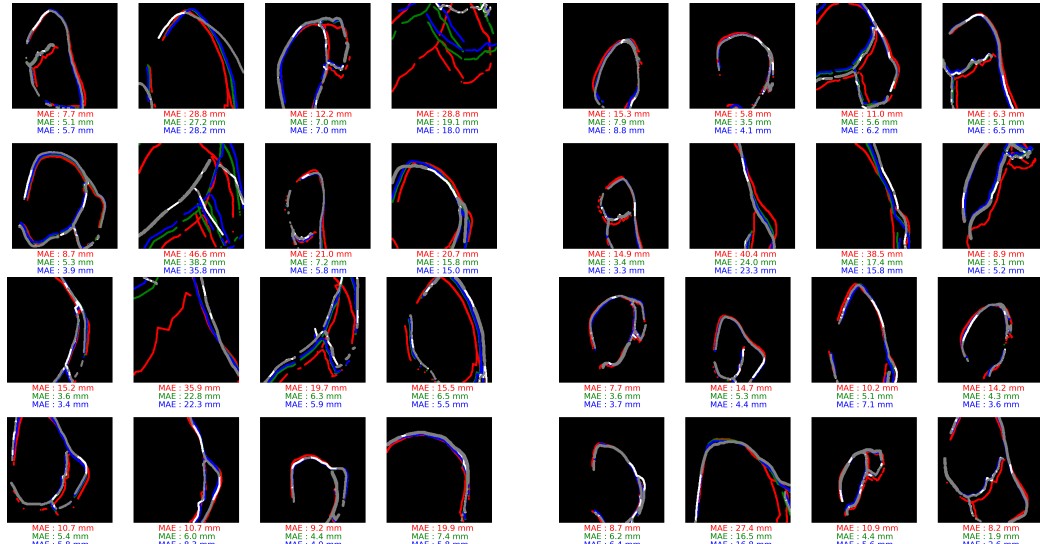

Figure 8: Results on the test dataset for 32 samples of Patient 1, using OLA. In gray, the initial segmentation mask, in white the projected GT primitives and in red, green and blue the respective projected results of each IEF iteration with associated MAE values.

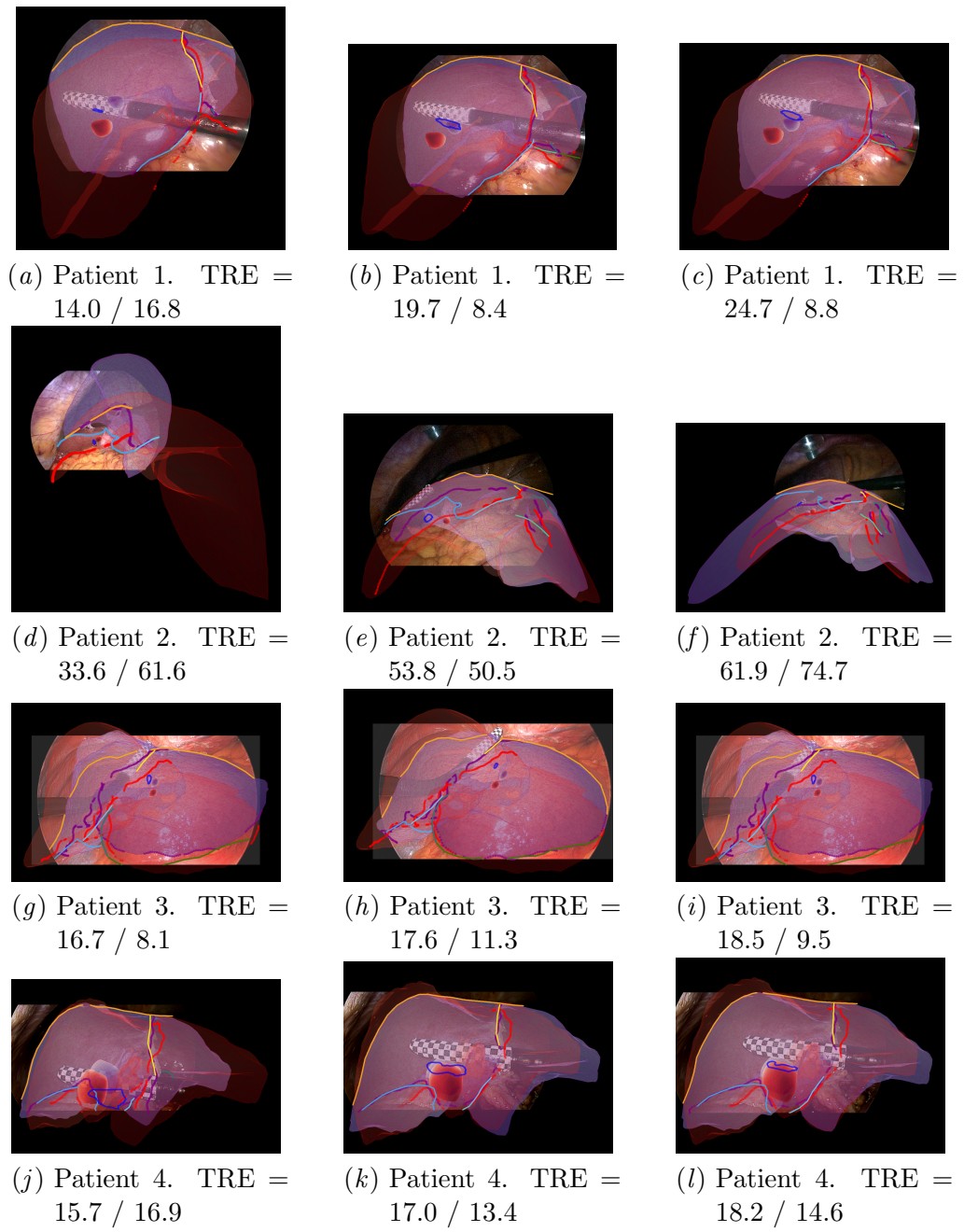

(a) Patient 1. TRE = 14.0 / 16.8

(b) Patient 1. TRE = 19.7 / 8.4

(c) Patient 1. TRE = 24.7 / 8.8

(d) Patient 2. TRE = 33.6 / 61.6

(e) Patient 2. TRE = 53.8 / 50.5

(f) Patient 2. TRE = 61.9 / 74.7

(g) Patient 3. TRE = 16.7 / 8.1

(h) Patient 3. TRE = 17.6 / 11.3

(i) Patient 3. TRE = 18.5 / 9.5

(j) Patient 4. TRE = 15.7 / 16.9

(k) Patient 4. TRE = 17.0 / 13.4

(l) Patient 4. TRE = 18.2 / 14.6

Figure 9: Comparative results between the Opt-B baseline and the proposed LMR-OLA method, for manual annotation of the primitives. Three images of each patient, selected as the minimal (left) and maximal (right) TRE, and the closest TRE to the mean (middle) of the LMR-OLA method. The TRE given in the captions are respectively for LMR-OLA / Opt-B. Opt-B results are in purple and LMR results are in red. Visible model landmarks are additionally shown using the same colour code. The projected tumour outline registered from ultrasound is in blue.

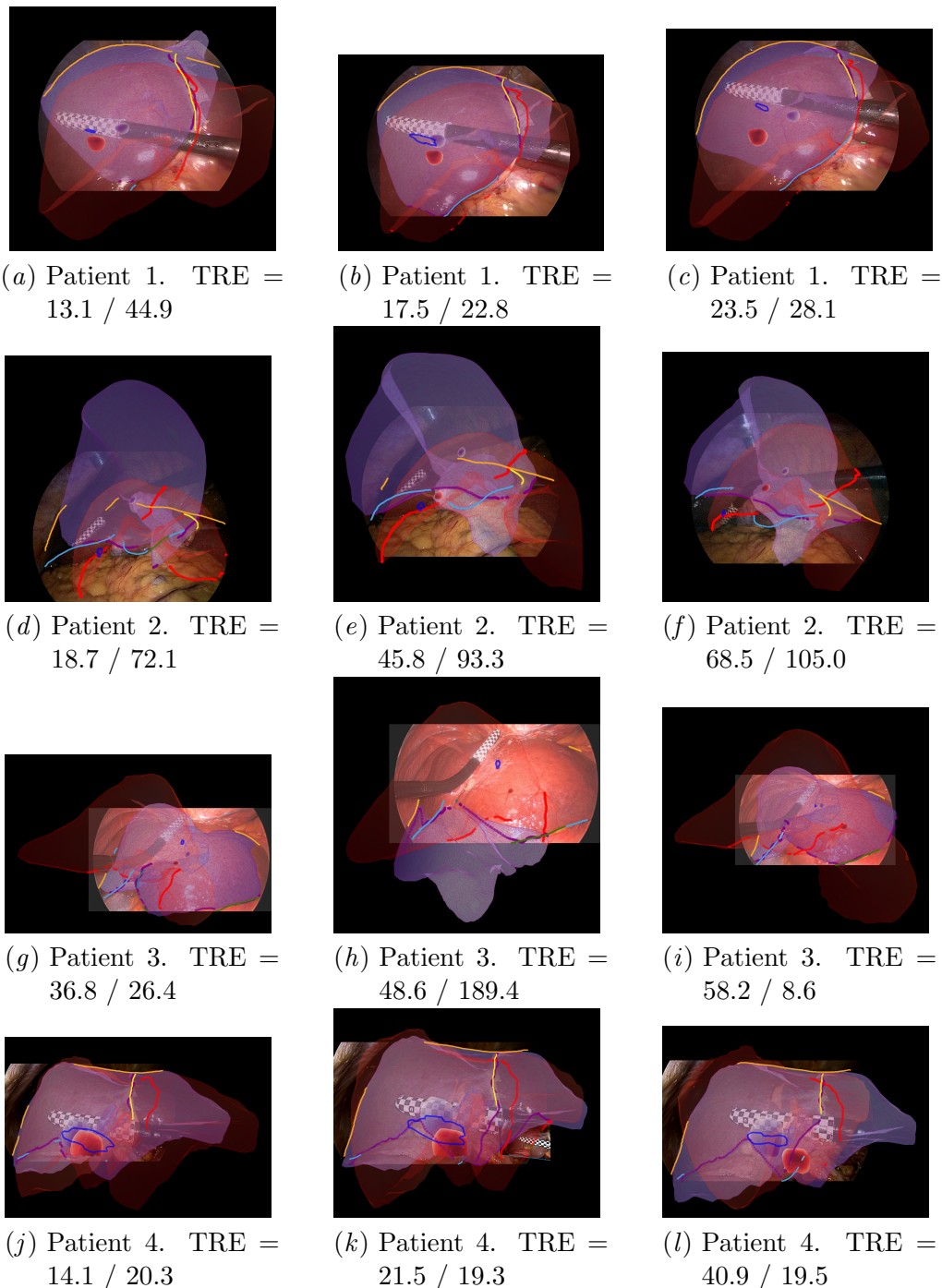

(a) Patient 1. TRE = 13.1 / 44.9

(b) Patient 1. TRE = 17.5 / 22.8

(c) Patient 1. TRE = 23.5 / 28.1

(d) Patient 2. TRE = 18.7 / 72.1

(e) Patient 2. TRE = 45.8 / 93.3

(f) Patient 2. TRE = 68.5 / 105.0

(g) Patient 3. TRE = 36.8 / 26.4

(h) Patient 3. TRE = 48.6 / 189.4

(i) Patient 3. TRE = 58.2 / 8.6

(j) Patient 4. TRE = 14.1 / 20.3

(k) Patient 4. TRE = 21.5 / 19.3

(l) Patient 4. TRE = 40.9 / 19.5

Figure 10: Comparative results between the Opt-B baseline and the proposed LMR-OLA method, for automatically detected primitives. The same structure and colour code as in figure 9 are used.

