# OpenReview forum: "Automatic 3D/2D Deformable Registration in Minimally Invasive Liver Resection using a Mesh Recovery Network"
_MIDL.io/2023/Conference — MIDL 2023 Poster_

### Official Review · Reviewer_imwY · 2023-02-03

**Confidence:** 2
**Preliminary Rating:** 3
**Recommendation:** Poster

**Summary:**

This paper presents a patient-specific registration pipeline for minimally invasive liver resection. The main contribution lies in the mesh recovery network and a strategy to generate synthetic patient-specific training data. The method is evaluated on a publicly available dataset and the experiments show promising results. The TRE is reported to be in an acceptable (comparable to state-of-the-art) range while being an order of magnitude faster.

**Strengths:**

- The introduction is well-written and gives a good overview about the topic and related work.
- The paper tackle the important problem of CAD for real-time AR for minimally invasive liver resection.
- The experiments show very promising results
- The authors present several figures in the appendix to visualize the results flow diagrams. I would recommend writing a more detailed explanation in the caption what the reader should learn from this figure.


**Weaknesses:**

- The author references a lot of other works which they are using in parts in their method. Without knowing all that work, it is very hard to understand the method.
- In general, the methods part is very hard to follow.
- It is not clear to me how the semi-synthetic dataset is generated.
- The test dataset only consists of 4 patients.
- The discussion including the limitations of the presented method is completely missing.


**Deanonymize Review:**

no

**Detailed Comments:**

- What is “Delaunay tetrahedralisation.”? Please give a reference.
- What modality to I have preoperatively? CT or MR? Why aren’t you using automatic segmentation approaches like TotalSegmentor to produce a initial liver segmentation which only needs to be corrected in some cases (Especially for the liver the automatic segmentation result should be almost perfect)
- What is meant with mobilization in this sentence: “Breathing was shown to have a limited influence on the deformation compared to mobilization” ?
- “We remove the first 6 deformation components in OHA, which essentially represent rigid displacements and a deformation induced by the fixed vena cava.” Why?


**Paper Type:**

both

**Questions To Address In The Rebuttal:**

Please rework the method section and try to make it easier to follow. I understand that the method is complex, and you need to present a lot of details. Maybe you could focus on the most important points and move details into the appendix.
Furthermore, I would like to see a fair discussion including limitations of the proposed work.

---

### Official Review · Reviewer_Bcd1 · 2023-02-03

**Confidence:** 2
**Preliminary Rating:** 4
**Recommendation:** Poster

**Summary:**

The paper presents a method for 3D/2D registration of the liver gland between a mesh model (typically segmented from a CT scan) and RGB images from an invasive camera during a laparoscopic procedure. The approach is based on a segmentation of the liver silhouette, which is then used as an input for the LMR (Liver Mesh Recovery) network. This network is inspired by the HMR (Human Mesh Recovery) framework and predicts the pose and deformation of the mesh model so that its projected contours match the ones from the image.
The fact that the registration is performedn by a neural network greatly speeds up the method and makes it the first one that could work in real time.

**Strengths:**

* This is really a system paper, where the authors combined existing technologies to solve an engineering challenge (i.e. make it run in real time). I think IPCAI/CARS would have been a more suited conference but on the other hand it is also not off-topic for MIDL. Furthermore, the idea of applying the Human Mesh Recovery method could be useful for other applications.
* The method makes sense, and achieves the goal of real-time.

**Weaknesses:**

* Only four patients were considered, which makes drawing conclusions about the applicability of the method harder.
* The errors with the automatic image annotations are significantly higher than the manual ones (sometimes higher than the previous MLR method). This means that even if the pipeline shows potential (in the sense that the contour information exists and can be used for registration), it is still a prototype and the contour segmentation problem still has to be solved.
* The part about using the COSNet network with co-attention but feeding the same image is a bit weird.

**Deanonymize Review:**

no

**Detailed Comments:**

I am not an expert in laparoscopy, so there is unfortunately so much I can comment on the application side of the paper (for instance assessing the final accuracy), but here are some comments/questions:

* Can we really take for granted that the pose of the liver can be unambiguously retrieved from the contours? I feel like this is a hypothesis that has not been discussed.
* A bit more details on the HMR network wouls have been useful (this is after all the core of the paper).
* Would there
* The semi-synthetic dataset is not well described: I had to look up the figures in the appendix to understand what it consists of.
* Figure 1 provides a overview of the pipeline, which is very important given its number of parts. However, I still find it a bit confusing: just looking at it was not enough to understand the pipeline and I had to read the text.
* The authors employ twice a big-O notation: `O(100)` and `O(n) = 10000`. While I think I understand what the authors mean, I would remove those notations because they arer wrongly employed (note that `O(100) = O(1) = O(1000000)`, or that `O(n) = 10000` does not mean anything mathematically).
* How was the M-Pose defined in practice, i.e. what was the protocol? How reliable/accurate is it?
* The methold yields very high errors on Patient A (due to large deformations according to the authors). Why wasn't the network then trained with larger deformations?


**Paper Type:**

validation/application paper

**Questions To Address In The Rebuttal:**

The paper is a bit difficult to follow for readers that are not expert of this particular field. I feel like the description of the pipeline could be improved.
My other questions are in the previous sections.

---

### Official Review · Reviewer_suhK · 2023-02-06

**Confidence:** 3
**Preliminary Rating:** 4
**Recommendation:** Oral

**Summary:**

The authors propose the Liver Mesh Recovery (LMR) framework.
It is a novel learning-based approach to registration in Minimally Invasive Liver Resection (MILR).
LMR is inspired by recent methods that reconstruct the human body as a 3D mesh from a single image.
The major contribution of the paper is the adaptation of such approaches to augment, in real-time, a patient-specific shape model of the liver.



**Strengths:**

The paper is well-written and the target application is clearly explained.
The literature review is appropriate and comprehensive.
The major contribution is the idea to train the model preoperatively from patient-specific semi-synthetic data. It enables the application of the approach in a real-time setting during surgery.
This paper provides several new contributions to the existing HMR approach.
Overall, the idea for the approach is interesting and the implementation and experiments are adequate.

**Weaknesses:**

The approach is interesting and has a lot of potential.
Further evaluation with real clinical data is required.
Other approaches for pose estimation should be considered. A comparison with more recent methods for pose estimation such as 'End-to-End Human Pose and Mesh Reconstruction with Transformers - Lin et al.' should be considered.
TRE of 2cm is significant.
The ultrasound probe should be considered in the analysis as this was the large source of error for patient 2.


**Deanonymize Review:**

no

**Paper Type:**

both

**Questions To Address In The Rebuttal:**

1. How are the weights used in the loss functions chosen? The learning rate schedule also seems quite specific. Please elaborate on how these parameters are determined.
2. In section 3.1.
"2) LMR uses the pinhole camera model, which reduces depth ambiguities, in contrast to the scaled-orthographic model in LMR" Do the authors mean HMR?
3. Could the area of the vena cava deform during surgery? Is not clear why it needs to be excluded from the FE analysis.
We have proposed LMR, a novel patient-specific learning-based registration method in

---

### Official Review · Reviewer_s1Fo · 2023-02-06

**Confidence:** 2
**Preliminary Rating:** 2
**Recommendation:** Poster

**Summary:**

The paper proposes a patient-specific liver mesh recovery (LMR) by using a learning-based method in order to solve the registration in minimally invasive liver resection preoperatively.
The main advantage of this paper in comparison to the SOTA is that the computational load in LMR occurs at training time allowing to run in real-time intraoperatively.

**Strengths:**

The main advantage of this paper in comparison to the SOTA is that the computational load in LMR occurs at training time allowing to run in real-time intraoperatively.
Another nice idea is the use of a physics-based rendering method to generate patient-specific image masks for a range of pose and deformation.

**Weaknesses:**

It would be nice if the methods and parameters chosen for the paper were further motivated and explained. In particular, on step 2, when the FEA model is described, there are many parameters that are described and assumptions that are made and an explanation for these choices would be great. For example, why the vertices of the vena cava are excluded. Another thing in this part is the Figure 6 which is very difficult to read. The color blue is somehow not visible and the plots of the liver are very small for the reader.
On the step 3, there are many choices as well on the rotation angles, and translations that are also not justified. The split between training and validation seems also quite small for the validation and test set and none cross-validation is mentioned.
The definition of the loss makes sense but the weights are as well not justified and the experiments with different parameters are not reported.
The authors claim that LMR's accuracy is on par with the state-of-the-art but only the values of the paper are reported without comparing to other registration papers and their outcomes. In particular, 2cm is reported as being a good accuracy for the liver registration.

**Deanonymize Review:**

no

**Paper Type:**

methodological development

**Questions To Address In The Rebuttal:**

1. What motivated the choices made for the methods and parameters in the paper, particularly in the FEA model described in step 2?
2. Can the authors provide more explanation for the exclusion of the vertices of the vena cava in the FEA model?
3. Can the authors improve the readability of Figure 6, as the blue color is difficult to see and the plots of the liver are small for the reader?
4. What was the justification for the choices made on the rotation angles and translations in step 3?
5. Can the authors provide more information on the split between the training and validation set and explain why cross-validation was not mentioned?
6. Can the authors justify the weights used in the loss definition and report on experiments with different parameters?
7. Can the authors provide comparison to other registration papers and their outcomes to support the claim that LMR's accuracy is on par with the state-of-the-art?
8. What led the authors to consider 2cm to be a good accuracy for liver registration?

---

### Official Review · Reviewer_C34J · 2023-02-07

**Confidence:** 4
**Preliminary Rating:** 3
**Recommendation:** Poster

**Summary:**

The authors propose the "Liver Mesh Recovery (LMR)" framework as a novel, deep learning-based registration method for Minimally Invasive Liver Resection (MILR) surgery. The majority of the computation load is shifted to the preoperative stage, which speeds up the computation time in the intraoperative image and enables offline training of patient-specific models.
The method is inspired by the recent Human Mesh Recovery framework and is the first learning-based method to solve registration in MILR. The authors evaluated the LMR method on a publicly available clinical dataset, both in semi-synthetic and real data, and achieved competitive results with low TRE and MAE in real clinical cases.  The method has been shown to be on par with state-of-the-art optimization-based approaches and is faster, with a speed up of five orders of magnitude. The pipeline could be improved with better accuracy from a larger dataset and improved physics-based simulation conditions.


**Strengths:**

1. The proposed learning-based registration method, LMR, offers a timely solution to the computational-intensive problem in medical image-based liver registration (MILR). The LMR method can speed up the intraoperative registration process and generate patient-specific models with fine-tuning in the preoperative stage. This is a significant improvement over traditional optimization-based methods.
2. The authors evaluated LMR using a publicly available clinical dataset, making the implementation more accessible and potentially more adaptable to real-world applications. The dataset includes real-patient cases, annotated preoperative and intraoperative images, camera calibration, and the ability to measure registration accuracy.
3. The use of whole body reconstruction in the registration process is a novel approach that has not been explored in MILR previously. This could potentially lead to further research and advancements in this field.

Overall, the proposed LMR method demonstrates a promising solution to the challenges faced in MILR and holds potential for real-world clinical applications


**Weaknesses:**

1. There is a lack of ablation study to identify the key components of the method and to understand the impact of network parameter selection on the performance of the method.
2. The experimental evaluation is limited to a single publicly available clinical dataset, so it would be desirable to validate the method on a larger and more diverse dataset to generalize the results and ensure robustness and accuracy.
3. The comparison experiments are limited to only optimization-based methods, and there is no comparison with other state-of-the-art registration methods, which could provide further insights into the performance of the proposed method.
4. The proposed method is based on learning-based techniques, which may have limited generalizability to unseen data and could be sensitive to outliers or noise in the data.

**Deanonymize Review:**

no

**Detailed Comments:**

1. The annotations for the Figure 1. is weak and hard to interpolate.

**Paper Type:**

both

**Questions To Address In The Rebuttal:**

1. To address the concern of unclear figure annotations, the authors could provide additional descriptions or captions to better clarify the results presented in Figure 1 and 2. This would help readers understand the significance of the results and follow the logic of the proposed method.
2. To further validate the proposed method, it would be valuable to compare it with other learning-based registration methods and evaluate its performance in comparison. This would provide a more comprehensive evaluation of the proposed method and give readers a better understanding of its strengths and limitations.
3. An ablation study could provide valuable insights into the importance of various parameters in the proposed method. By systematically removing or changing different components, the authors could identify which factors contribute most to the performance of the method and provide evidence for the validity of their choices. This would also help to ensure that the results are robust and generalizable to other datasets.

---

### Meta-Review · Area_Chair_BWes · 2023-02-23

**Recommendation:** Accept (Poster)
**Confidence:** 4

**Metareview:**

This work proposes a learning-based registration framework for Minimally Invasive Liver Resection (MILR) surgery. Overall, this paper gives a clear introduction of clinical background and related works. The methodology is more clear after revision. The proposed method can be significant in clinical applications due to higher computational efficiency compared with conventional algorithms.

Pros:
- The learning-based framework is highly efficient with good novelty.
- Experimental results demonstrate its promising performance and applications

Cons:
- Validation is not sufficient due to lack of real-world datasets with labels
- The comparison with other learning-based registration methods (not limited to MILR) is needed

Overall, among five reviewers, two reviewers rated borderline, two reviewers rated weak accept, and one reviewer rated weak reject but with a low confidence. After sufficiently considering the pros of this work, I recommend to accept this paper.